# Aurora Kinase A and Bcl-xL Inhibition Suppresses Metastasis in Triple-Negative Breast Cancer

**DOI:** 10.3390/ijms231710053

**Published:** 2022-09-02

**Authors:** Natascha Skov, Carla L. Alves, Sidse Ehmsen, Henrik J. Ditzel

**Affiliations:** 1Department of Cancer and Inflammation Research, Institute of Molecular Medicine, University of Southern Denmark, 5000 Odense, Denmark; 2Department of Oncology, Institute of Clinical Research, Odense University Hospital, 5000 Odense, Denmark; 3Academy of Geriatric Cancer Research (AgeCare), Odense University Hospital, 5000 Odense, Denmark

**Keywords:** *AURKA*, *BCL2L1*, basal-like breast cancer, epithelial–mesenchymal transition, alisertib

## Abstract

Triple-negative breast cancer (TNBC) is a heterogeneous disease that accounts for 10–15% of all breast cancer cases. Within TNBC, the treatment of basal B is the most challenging due to its highly invasive potential, and thus treatments to suppress metastasis formation in this subgroup are urgently needed. However, the mechanisms underlying the metastatic ability of TNBC remain unclear. In the present study, we investigated the role of Aurora A and Bcl-xL in regulating basal B cell invasion. We found gene amplification and elevated protein expression in the basal B cells, which also showed increased invasiveness in vitro, compared to basal A cells. Chemical inhibition of Aurora A with alisertib and siRNA-mediated knockdown of *BCL2L1* decreased the number of invading cells compared to non-treated cells in basal B cell lines. The analysis of the correlation between *AURKA* and *BCL2L1* expression in TNBC and patient survival revealed significantly decreased relapse-free survival (*n* = 534, *p* = 0.012) and distant metastasis-free survival (*n* = 424, *p* = 0.017) in patients with primary tumors exhibiting a high combined expression of *AURKA* and *BCL2L1*. Together, our findings suggest that high levels of Aurora A and Bcl-xL promote metastasis, and inhibition of these proteins may suppress metastasis and improve patient survival in basal B TNBC.

## 1. Introduction

Breast cancer is the most common cancer in women worldwide, accounting for 31% of all cancer cases and is the second leading cause of cancer-related death at 15% of all breast cancer deaths [1]. Although prognosis for women diagnosed with localized disease is good with a 5-year survival rate of 87%, the 5-year survival rate for those diagnosed with distant metastasis is only 23.4% [2]. Of all breast cancer subtypes, the triple-negative breast cancer (TNBC), which accounts for 10–15% of all breast cancer cases, is associated with worse prognosis compared to other subtypes due to its aggressive nature, higher metastatic rate and early recurrence [3,4]. Treatment of this breast cancer subgroup is very challenging due to the lack of available treatments and tumor heterogeneity [5]. Although chemotherapy has been shown to improve TNBC patient survival, it has limited efficacy after the tumor has metastasized [6]. Importantly, several targeted therapies have recently been approved for TNBC. PARP (poly adenosine diphosphate-ribose polymerase) inhibitors have been approved for BRCA1 and BRCA2 mutation carriers, which account for approximately 20% of TNBC patients [7]. Additionally, an antibody conjugate consisting of an anti-Trop2 antibody conjugated with SN-38 (Sacituzumab govitecan), the active metabolite of irinotecan, was recently approved by the FDA for metastatic TNBC based on promising phase III study results [8,9]. Finally, the addition of the anti-PDL1 pembrolizumab to neoadjuvant chemotherapy has shown to significantly improve event-free survival for high-risk TNBC [10].

TNBC significantly overlaps with the molecular basal-like breast cancer subtype [11], which can be divided into: basal A, with mixed basal/luminal/epithelial-like features; and basal B, exhibiting mesenchymal-like features and spindle-like morphology [12,13]. The epithelial-like phenotype can, under normal circumstances, switch to the mesenchymal-like phenotype through epithelial–mesenchymal transition (EMT), and in cancer cells, this process has been shown to play a key role in cancer invasion and metastasis [14,15,16,17]. Recently, we showed that gene copy number variations (CNV) rather than specific mutations are the leading cause of breast cancer spread and lymph node (LN) metastasis [18]. Using single cell sequencing, we identified the CNV of chromosome 20q in LN metastasis compared to the primary tumor and found that amplification of *AURKA* (protein: Aurora Kinase A (Aurora A)) and *BCL2L1* (protein: Bcl-xL) was associated with LN metastasis. Furthermore, overexpression of Aurora A and Bcl-xL has been associated with breast cancer metastasis through the induction of EMT [19,20].

Aurora A is a mitosis regulator required for genome stability, which controls the G2/M transition via the phosphorylation of breast cancer type 1 (BRCA1) and activation of the cyclin-dependent kinase 1 (CDK1)/cyclin B complex [21]. Aurora A overexpression has been identified in many solid tumors, including lung, ovarian, colorectal and breast cancer, as well as in hematologic tumors such as T-cell lymphoma and is linked to poor prognosis, drug resistance and degradation of the tumor suppressor p53 [14,15,19,21,22,23]. Several Aurora A inhibitors have been developed, including alisertib (MLN8237), which are currently undergoing numerous phase 1 and 2 clinical trials in breast and other cancers [24]. Alisertib is a selective Aurora A inhibitor that prevents phosphorylation of Aurora A, leading to cell cycle arrest in the G2/M phase, autophagy and apoptosis, with the latter likely by altering the expression of B-cell lymphoma 2 (BCL-2) family members [25,26]. The BCL-2 protein family is characterized by the BCL-2 homology domains (BH1-4) and includes the anti-apoptotic BH1-4 protein Bcl-xL [27], which was found to also have other functions [28]. Importantly, metastasis may be enhanced by the ability of cancer cells to resist apoptosis, and highly metastatic cancer cells likely exhibit a greater survival ability and resistance to apoptosis than poorly metastatic cells [29,30,31]. Indeed, it has been shown that Bcl-xL expression increases the metastatic potential of breast cancer cells in vivo by promoting resistance to growth factors and organ-derived cytokines [32]. Furthermore, Bcl-xL can improve cell survival and enhance anchorage-independent growth, which may cause metastasis [32]. Many inhibitors targeting the BCL-2 family proteins have been developed, but Bcl-xL specific inhibitors are not yet available.

In this study analyzing TNBC cell lines, we investigated the role of Aurora A and Bcl-xL amplification and overexpression in two mesenchymal-like basal B cell lines compared to an epithelial-like basal A breast cancer cell line. Higher copy number and expression of Aurora A and Bcl-xL were associated with higher invasion ability in basal B cell lines. Furthermore, we observed that Aurora A inhibition with alisertib and siRNA-mediated *BCL2L1* knockdown decreased the invasion of basal B cells. Finally, we showed that clinical TNBC samples co-expressing higher mRNA levels of *AURKA* and *BCL2L1* correlated with shorter relapse-free survival (RFS) and distant metastasis-free survival (DMFS) of patients with TNBC. Our results suggest that targeting Aurora A and Bcl-xL to suppress tumor metastasis in the basal B TNBC subgroup, which has limited therapeutic options, may be a useful therapeutic strategy.

## 2. Results

### 2.1. TNBC Cells with the Basal B-Like Phenotype Exhibit Gene Amplification and Overexpression of Aurora A and Bcl-xL

Although Aurora A and Bcl-xL have both been linked to cancer formation and metastasis in breast cancer [19,32], their specific role in the ability of basal A and B cells to invade and metastasize remains unclear. To examine the correlation between Aurora A and Bcl-xL expression and metastatic abilities in basal A and basal B TNBC, we first confirmed the epithelial or mesenchymal features of MDA-MB-468, CAL-120 and MDA-MB-231 cells (Figure 1a) [33] and evaluated their invasion ability in vitro (Figure 1b). As expected, immunocytochemical analysis using the mesenchymal marker, vimentin and the epithelial marker, EPCAM showed: low vimentin and high EPCAM in MDA-MB-468 cells, consistent with epithelial basal A phenotype; and high-vimentin and low-EPCAM expression in CAL-120 and MDA-MB-231 cells, consistent with a mesenchymal basal B phenotype (Figure 1a). An invasion assay showed that CAL-120 and MDA-MB-231 cells exhibited higher invasive properties than MDA-MB-468 cell lines (Figure 1b). Although the higher invasion ability of CAL-120 cells compared to MDA-MB-468 cells did not reach statistical significance, the differences observed between these two cell lines in vimentin and EPCAM expression support the mesenchymal phenotype of CAL-120 cells. These findings support the classification of MDA-MB-468 cells as basal A subtype and CAL-120 and MDA-MB-231 cells as basal B subtype, and that basal B exhibits higher invasive properties.

Next, we evaluated *AURKA* and *BCL2L1* copy numbers and Aurora A and Bcl-xL protein levels in the two basal B (CAL-120 and MDA-MB-231) cell lines compared to the basal A cell line (MDA-MB-468). *AURKA* and *BCL2L1* amplification were observed in both basal B cell lines when compared to the basal A cell line (Figure 2a). Importantly, examining protein levels by Western blotting showed higher expression of both Aurora A and Bcl-xL in the basal B cell lines than in the basal A cells (Figure 2b), with CAL-120 exhibiting the highest Aurora A expression, while MDA-MB-231 showed the highest Bcl-xL expression. Comparable results were obtained with immunocytochemical analysis of these proteins in the three cell lines (Figure 2c). Together, these findings show that TNBC cell lines with the basal B phenotype (basal B TNBC) are more invasive and exhibit higher levels of both Aurora A and Bcl-xL.

### 2.2. Inhibition of Aurora A and BCL2L1 Reduces Invasion of Basal B TNBC Cell Lines

To further investigate the role of Aurora A in the invasion capability of basal B TNBC cells, we tested a specific Aurora A inhibitor, alisertib and evaluated its effect on cell viability and invasion (Figure 3). Alisertib IC50 value for CAL120 and MDA-MB-231 cells was determined as 10 and 19.33 µM, respectively, and 20 µM alisertib was used for further experiments in both cell lines (Figure 3a). Furthermore, alisertib induced an increased Aurora A and Bcl-xL expression and decreased vimentin compared to untreated CAL-120 and MDA-MB-231 cells as evaluated by Western blotting (Figure 3b). Importantly, treatment with alisertib inhibited CAL-120 and MDA-MB-231 cell invasiveness (Figure 3c), consistent with reduced levels of vimentin observed upon treatment (Figure 3b). Although alisertib reduced invasion in MDA-MB-468 cells as well, vimentin levels in this cell line were significantly lower (Figure 3b) which supports the lower invasiveness and aggressiveness of these cells. Together, these findings suggest an important role for Aurora A in regulating invasion in highly aggressive basal B TNBC cells.

As Bcl-xL expression was found to be significantly higher in the more invasive basal B cells compared to the less invasive basal A cells (Figure 2b,c and Figure 3b), we further investigated the role of *BCL2L1* in the invasion capability of basal B TNBC cells. To this end, we performed siRNA-mediated transient knockdown of *BCL2L1* and evaluated its effect on cell growth and invasion (Figure 4). Bcl-xL expression was significantly reduced at 72 h in both cell lines treated with *BCL2L1*–siRNA compared to cells treated with scrambled siRNA, as assessed by Western blotting (Figure 4a). We observed decreased cell growth of CAL-120 and MDA-MB-231 cells 72 h following *BCL2L1*–siRNA transfection compared to cells transfected with the control siRNA, as evaluated by crystal violet assay (Figure 4b). Notably, *BCL2L1* siRNA-mediated knockdown caused a marked reduction in the invasiveness of CAL-120 and MDA-MB-231 cells (Figure 4c). Together, these findings support a role for *BCL2L1* in controlling basal B TNBC cells invasion. A schematic presentation of the role of Aurora A and Bcl-xL in regulating the invasion of basal B cells is shown in Figure 5.

### 2.3. High AURKA and BCL2L1 Expression Correlates with Poor Clinical Outcome in TNBC

To investigate the prognostic value of Aurora A and Bcl-xL in cohorts of TNBC patients, we used the web-based tool Kaplan–Meier plotter to access the correlation between *AURKA* and *BCL2L1* mRNA expression and clinical outcome. High *BCL2L1* expression significantly correlated with shorter relapse-free survival (RFS, *n* = 534, *p* = 0.0019) and distant metastasis-free survival (DMFS, *n* = 424, *p* = 0.0055) in TNBC patients (Figure 6a). High *AURKA* expression was also associated with shorter RFS and DMFS in the same patient population, but this correlation was not statistically significant (Figure 6b, *p* = 0.017 and *p* = 0.13, respectively). Notably, mean expression of combined *AURKA* and *BCL2L1* was calculated, and high *AURKA*-*BCL2L1* mRNA expression showed a significant correlation with low RFS (*p* = 0.012) and DMFS (*p* = 0.017) (Figure 6c). Cox regression multivariate analysis including mean expression of combined *AURKA* and *BCL2L1*, *MKi67* and *ESR1* expression showed that combined *AURKA* and *BCL2L1* and *ESR1* expression are independent prognostic factors of RFS (HR 1.65, *p* = 0.0036; HR 1.42, *p* = 0.041, respectively) and DMFS (HR 1.74, *p* = 0.013; HR 1.49, *p* = 0.041, respectively; Table 1.) The KM plotter does not subdivide the basal-like subtypes into basal A and B; thus, we were not able to perform survival analysis of these markers separately in basal A and basal B TNBC. These data support a correlation between high *AURKA* and *BCL2L1* and poor clinical outcome in TNBC patients.

## 3. Discussion

TNBC and the partly overlapping molecular subtype basal-like breast cancer represent an aggressive subtype of breast cancer that is associated with high metastatic ability, tumor heterogeneity and lack of effective therapies [4,34]. Although chemotherapy is initially effective, the majority of TNBC, particularly the basal B subtype, will metastasize and exhibit chemotherapy resistance, which limits the efficacy of further therapy and is associated with poor clinical outcomes [35,36,37]. Investigating the mechanisms of metastasis formation is essential for developing effective clinical interventions for this patient population [38]. Aurora A and Bcl-xL have both been linked to the metastatic process in different types of cancer, including breast cancer [19,32], but their specific impact on the ability of TNBC cells to invade and metastasize remains unclear.

Here, we showed gene amplification and protein upregulation of Aurora A and Bcl-xL in basal B TNBC cell lines compared to basal A cells, which correlated with a mesenchymal phenotype and higher invasive capabilities of basal B cells. Other studies have found upregulation of Aurora A and Bcl-xL in breast cancer, including basal-like breast cancer and an association with breast cancer progression through the activation of EMT and tumor stemness [19,20]. However, concomitant high expression of these two molecules has not yet been shown in basal B TNBC. Identifying the vulnerabilities of TNBC is particularly challenging due to its high heterogeneity and lack of major cancer drivers. Thus, identification of the co-expression of Bcl-xL and Aurora A opens the possibility for co-targeting therapeutic strategies in TNBC. Indeed, targeting the anti-apoptotic protein Bcl-xL together with anti-mitotic agents has recently been proposed as an efficient therapeutic strategy for different cancers, including TNBC [39]. In a different study, it was shown that TNBC cells expressing Bcl-xL are sensitive to combined inhibition with a drug targeting Bcl-xL and a drug targeting CDK1/2/4 but not when the Bcl-xL inhibitor was combined with drugs inhibiting FOXM1, CDK4/6, Aurora A or Aurora B [40]. In contrast, our data show that basal B TNBC cell lines co-expressing Aurora A and Bcl-xL are sensitive to Aurora A inhibition with alisertib. Furthermore, we found that treatment with alisertib induces increased expression of Bcl-xL in basal B TNBC cells, further supporting co-targeting both molecules. In line with our findings, it was recently shown that the combined inhibition of Aurora B and Bcl-xL caused synergistic cell growth impairment in cancer cells [41,42].

The Aurora A inhibitor alisertib has been shown to hinder the phosphorylation of Aurora A and induce apoptosis by altering the expression of the BCL-2 family proteins to a pro-apoptotic state through increased expression of pro-apoptotic proteins and decreased expression of anti-apoptotic proteins, such as Bcl-xL [25]. In our study, we observed an increased Aurora A expression after treatment with alisertib in basal B TNBC cells, which can be explained by the accumulation of non-phosphorylated Aurora A. Furthermore, we showed that the reduced metabolic activity induced by alisertib in basal B cells is not due to the induction of apoptosis, as Bcl-xL expression is increased, likely due to a compensation mechanism in response to the growth inhibition caused by the Aurora A inhibitor. Instead, the reduced cell viability can be a result of the cell cycle arrest caused by Aurora A inactivation. Importantly, inhibition of Aurora A and knockdown of *BCL2L1* resulted in reduced in vitro invasion of both basal B cells compared to non-treated cells, suggesting that these two targets are important in the metastatic process in mesenchymal-like TNBC.

To further investigate the implications of overexpression of Aurora A and Bcl-xL in TNBC breast cancer patients, we assessed the correlation between *AURKA* and *BCL2L1* mRNA expression and clinical outcomes using the web-based tool KM plotter. We found that high-*BCL2L1*-*AURKA* correlated with shorter RFS and DMFS. High levels of Bcl-xL or Aurora A have been previously correlated with poor overall survival (OS) and short progression-free survival (PFS) in breast cancer, including TNBC [40,43,44,45,46], but a correlation between co-expression of *AURKA* and *BCL2L1* and TNBC patient survival has not yet been reported. Of the studies that evaluated these markers and clinical outcomes in TNBC, one study found a correlation between *AURKA* and OS and PFS in a small population of TNBC patients (*n* = 122) [44]. Another study showed an association between *BCL2L1* levels and OS in a larger TNBC patient population (*n* = 580), but only one microarray dataset was used, and RFS and DMFS were not evaluated [40]. Furthermore, this study showed that high levels of *AURKA* either alone or together with *BCL2L1* levels were not associated with altered patient survival [40]. Our study is the first, to our knowledge, to show that high co-expression of *AURKA* and *BCL2L1* correlates with poor RFS and DMFS in a large population of TNBC patients from different datasets.

In summary, our evaluation of the association between Aurora A and Bcl-xL copy numbers and expression and epithelial/mesenchymal phenotypes in TNBC shows that the co-expression of Aurora A and Bcl-xL is associated with the invasion capability of TNBC cells in the basal B subtype, and inhibition of both molecules is required to suppress tumor metastasis. Finally, we suggest that combined *AURKA* and *BCL2L1* expression is a prognostic factor in TNBC as tumors exhibiting high expression of both markers correlate with poor outcomes.

## 4. Materials and Methods

### 4.1. Cell Line Culture

The TNBC cell lines MDA-MB-468 and MDA-MB-231 were obtained from ATCC, and CAL-120 was obtained from Leibniz Institute DSMZ—German Collection of Microorganisms and Cell Cultures. All cell lines were grown in Dulbecco’s Modified Eagle Medium (DMEM, Sigma, Darmstadt, Germany) supplemented with 10% fetal bovine serum (FBS, Gibco, ThermoFisher Scientific, Waltham, MA, USA) and 1% penicillin/streptomycin (P/S, Gibco, ThermoFisher Scientific, Waltham, MA, USA). All cell lines underwent DNA authentication using Cell ID™ System (Promega, Walldorf, Germany) and mycoplasma testing (Lonza, Basel, Switzerland) before use in the described experiments.

### 4.2. DNA Extraction and CNV Assay

DNA was extracted using the DNeasy Blood and Tissue kit (Qiagen, Hilden, Germany) according to manufacturer instructions. The DNA was diluted in 1× tris-EDTA (TE) buffer to a concentration of 5 ng/L before running TaqMan Copy Number Assay PCR (Applied Biosystems, ThermoFisher Scientific, Waltham, MA, USA) on a StepOnePlus Real-Time PCR system v2.3 (Applied Biosystems, ThermoFisher Scientific, Waltham, MA, USA) in four replicates using *AURKA* (Hs02938272_cn, Applied Biosystems, ThermoFisher Scientific, Waltham, MA, USA) and *BCL2L1* (Hs07178628_cn, Applied Biosystems, ThermoFisher Scientific, Waltham, MA, USA) copy number assays. TaqMan Copy Number Reference Assay RNase P (4403326, ThermoFisher Scientific, Waltham, MA, USA) was used as a control. The threshold was set to 0.2 for all targets, and data were analyzed with CopyCaller Software v2.1 (Applied Biosystems, ThermoFisher Scientific, Waltham, MA, USA).

### 4.3. Protein Extraction and Western Blot

Cells were lysed in RIPA buffer (10mM Tris HCL (pH 8), 5mM Na_2_EDTA (pH 8), 1% NP-40, 0.5% sodium deoxycholate, 0.1% sodium dodecyl-sulfate (SDS)) containing protease and phosphatase inhibitors (Complete Mini and PhosStop, Sigma, Darmstadt, Germany ). Total protein was quantified using Pierce bicinchoninic acid (BCA) protein assay (ThermoFisher Scientific, Waltham, MA, USA). SDS-PAGE was run with 15 μg protein lysate in each well. The protein was transferred from the gel to a polyvinylidene fluoride (PVDF) transfer membrane and incubated with primary anti-Aurora A antibody (HPA002636, Sigma, Darmstadt, Germany, dilution 1:250), anti-Bcl-xL antibody (2762S, Cell Signaling Technology, Frankfurt, Germany, dilution 1:1000), anti-vimentin antibody (V6630, Sigma, Darmstadt, Germany, dilution 1:1000) and HRP-conjugated secondary antibody (anti-rabbit or anti-mouse IgG, P0448, P0447, Dako, Glostrup, Denmark, dilution 1:5000). Anti-GAPDH antibody (clone 6C5, Santa Cruz Biotechnology, Dallas, TX, USA, dilution 1:20,000) was used as loading control. The membrane was developed using enhanced chemiluminescence (ECL) Western blot kit (GE Healthcare, Buckinghamshire, UK) and visualized on Fusion-Fx7-7026 WL/26MX instrument (Vilber, Collégien, France).

### 4.4. Immunocytochemistry

Sections of 4 µm were cut from formalin-fixed and paraffin-embedded (FFPE) blocks of cell lines and incubated with anti-Aurora A (HPA002636, Sigma, Darmstadt, Germany, dilution 1: 250), anti-Bcl-xL (2762S, Cell Signaling Technology, Frankfurt, Germany, dilution 1: 1000), anti-vimentin (clone V9, Dako, Glostrup, Denmark, dilution 1:8000) and anti-EPCAM (clone Ber-EP4, Dako, Glostrup, Denmark, dilution 1:250) antibodies. Microscopy of the cells was performed on a Leica DMLB microscope (1003/numerical aperture [NA] 1.25; Leica Microsystems, Wetzlar, Germany) with 40× objectives using LasV3.6 acquisition software.

### 4.5. Aurora A Inhibition with Alisertib

Alisertib (name MLN8237, formula C_27_H_20_ClFN_4_O_4_, #HY-10971, Medchem Express, Sollentuna, Sweden), an orally active and selective Aurora A kinase inhibitor (IC50 = 1.2 nM), which binds to Aurora A kinase resulting in mitotic spindle abnormalities and mitotic accumulation, was dissolved in dimethyl sulfoxide (DMSO, Sigma, Darmstadt, Germany). The effect of alisertib on CAL-120 and MDA-MB-231 cells viability was determined by a dose–response curve based on different drug concentrations of alisertib ranging from 0.0001 µM to 20 µM, and DMSO as vehicle, for 72 h. Cell viability was measured using 3-(4,5-dimethylthiazol-2-yl)-2,5-diphenylte-trazolium bromide (MTT) assay.

### 4.6. BCL2L1 siRNA Knockdown

*BCL2L1* gene knockdown was performed using Silencer Validated siRNA (Sigma, Darmstadt, Germany) with the sequence: sense GGAGAACCACUACAUGCAGtt and antisense CUGCAUGUAGUGGUUCUCCtg on CAL-120 and MDA-MB-231 cells. Chemical transfection was done with Lipofectamine RNAiMAX reagent (Invitrogen, ThermoFisher Scientific, Waltham, MA, USA) in Opti-MEM media (Gibco, ThermoFisher Scientific, Waltham, MA, USA) and MISSION siRNA Universal Negative Control (SIC001, Sigma, Darmstadt, Germany). Efficiency was evaluated at 24, 48 and 72 h after transfection with Western blotting. The effect of siRNA-mediated knockdown on cells growth was evaluated with crystal violet colorimetric assay.

### 4.7. Cell Proliferation Assay

Transfected cells were seeded (10^4^ cells/well) in 96-well plates and incubated for 24, 48 and 96 h at 37 °C in 5% CO_2_ for evaluation of cell proliferation using crystal violet-based colorimetric assay [47]. The absorbance was measured in Sunrise™ 500 absorbance reader (Tecan, Männedorf, Switzerland).

### 4.8. Cell Invasion Assay

Cell invasion was measured using the cell invasion assay kit (ECM554, Sigma, Darmstadt, Germany) according to manufacturer instructions. The cells were starved in serum-free growth medium for 24 h before seeding 2.5 × 10^5^ cells per insert in a 24-well plate. The plate was incubated for 72 h at 37 °C and 5% CO_2_ humidity. For the inhibitor and siRNA-mediated knockdown experiments, cells were first treated with alisertib or *BCL2L1*-siRNA in medium containing serum for 48 h, after which the media was changed to serum-free growth medium with alisertib treatment. Next day, cells were seeded in inserts, and invasion was evaluated 24 h after seeding. Cell growth medium with serum was used as a chemoattractant. Cells without chemoattractant under the insert were used as a non-invasive control for the invasion assay. Cell invasion was measured with fluorescent light emission (480 nm/520 nm) using a Victor3™ 1420 counter (Perkin Elmer, Waltham, MA, USA).

### 4.9. Kaplan–Meier Plotting

The web tool Kaplan–Meier (KM) plotter (Budapest, Hungary) [48] was used to generate survival curves for TNBC patients depending on mRNA expression (gene chip) of *AURKA* and *BCL2L1*. All datasets available in KM plotter were included in the analysis. The inclusion criteria for sample selection were: ER and PR negative by IHC and HER2 negative by array, independently of clinical and pathological characteristics, such as grade, lymph node status or previous treatment. JetSet optimal probe was selected for each gene (probe Id 208,079 for *AURKA* and probe Id 212,312 for *BCL2L1*) and the best performing threshold was selected as the cut-off. Mean expression of both genes was used to evaluate the correlation of combined *AURKA* and *BCL2L1* expression. Relapse-free survival (RFS) and distant metastasis-free survival (DMFS) were selected as endpoints.

### 4.10. Statistical Analysis

Statistical significance was calculated using one-way ANOVA, and *p*-values ≤ 0.05 were considered statistically significant. GraphPad Prism v.8 (GraphPad Software, Inc., San Diego, CA, USA) was used for statistical analysis.

## Figures and Tables

**Figure 1 ijms-23-10053-f001:**
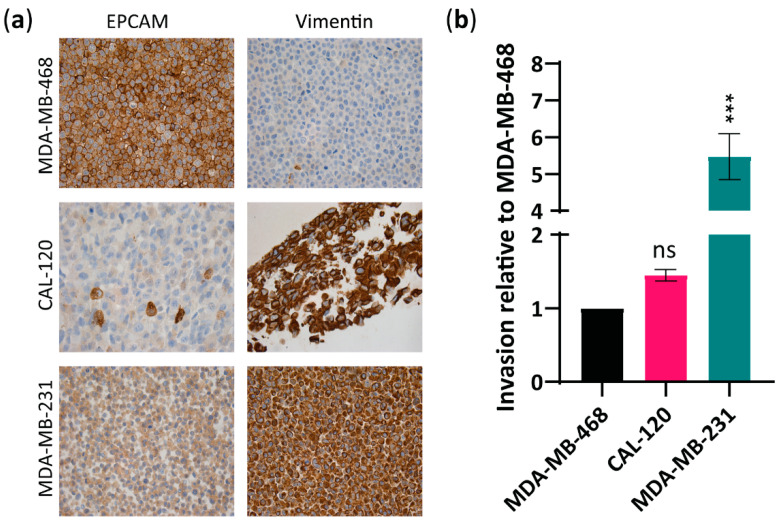
Basal B breast cancer cell lines CAL-120 and MDA-MB-231 showed mesenchymal phenotype and increased invasiveness versus MDA-MB-468 basal A cells exhibiting an epithelial phenotype. (**a**) Immunocytochemical analysis of formalin-fixed paraffin-embedded (FFPE) basal A, MDA-MB-468 cells and basal B, CAL-120 and MDA-MB-231 cells lines, stained for EPCAM and vimentin (×20); (**b**) Invasive properties of basal B cells relative to the basal A cell line as determined by a cell invasion assay kit. The data represent the mean of three biological replicates ± SEM. Statistically significant differences calculated by one-way ANOVA are shown as ns > 0.05 and *** *p* ≤ 0.001.

**Figure 2 ijms-23-10053-f002:**
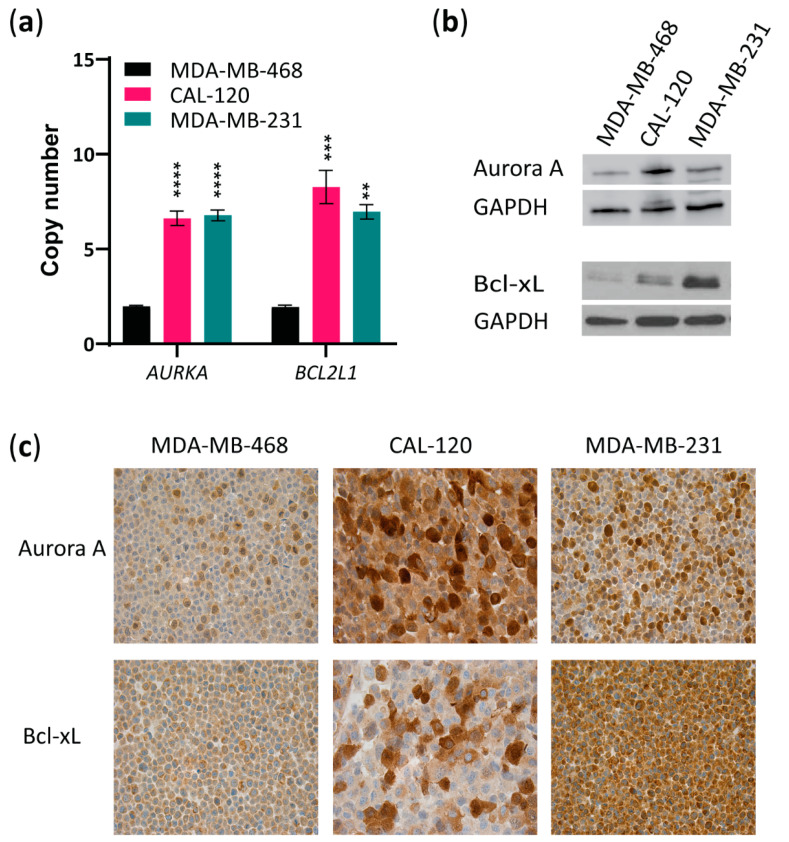
Basal B TNBC cells exhibit amplification of *AURKA* and *BCL2L1* and overexpression of Aurora A and Bcl-xL. (**a**) Evaluation of *AURKA* and *BCL2L1* gene copy numbers by quantitative RT-PCR copy number assay using TaqMan copy number variant (CNV) primers in basal A or B breast cancer cell lines. RNase P was used as reference. The data represent the mean of three biological replicates ± SEM. Statistically significant differences calculated by one-way ANOVA are shown as ns > 0.05, ** *p* ≤ 0.01, *** *p* ≤ 0.001 and **** *p* ≤ 0.0001; (**b**) Western blot analysis of Aurora A and Bcl-xL in lysates from MDA-MB-468, CAL-120 and MDA-MB-231 cell lines. GAPDH was used as loading control. A representative of 2 biological replicates is shown; (**c**) Immunocytochemical analysis of FFPE basal A, MDA-MB-468 cells and basal B, CAL-120 and MDA-MB-231 cell lines, stained for Aurora A and Bcl-xL (×20).

**Figure 3 ijms-23-10053-f003:**
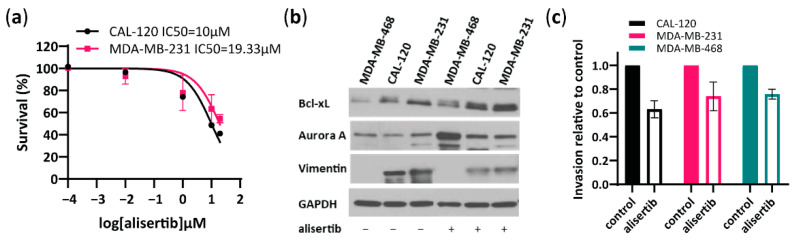
Aurora A inhibitor, alisertib reduces the invasive properties of basal B breast cancer cells. (**a**) Concentration-dependent inhibition of cell viability performed in basal B cells to determine the IC50 of Aurora A inhibitor alisertib, as assessed by an MTT assay 72 h after treatment; (**b**) Western blot analysis of Aurora A, Bcl-xL and vimentin in MDA-MB-468, CAL-120 and MDA-MB-231 cells before and after treatment with alisertib (10 µM in MDA-MB-468 cells and 20 µM in CAL-120 and MDA-MDA-MB-231 cells, treated for 72 h). GAPDH was used as loading control; (**c**) Analysis of invasion properties of basal A and basal B cells treated with alisertib (10 µM in MDA-MB-468 cells and 20 µM in CAL-120 and MDA-MB-231 cells, treated for 72 h) relative to cells without chemoattractant. The data represent the mean of two biological replicates ± SEM.

**Figure 4 ijms-23-10053-f004:**
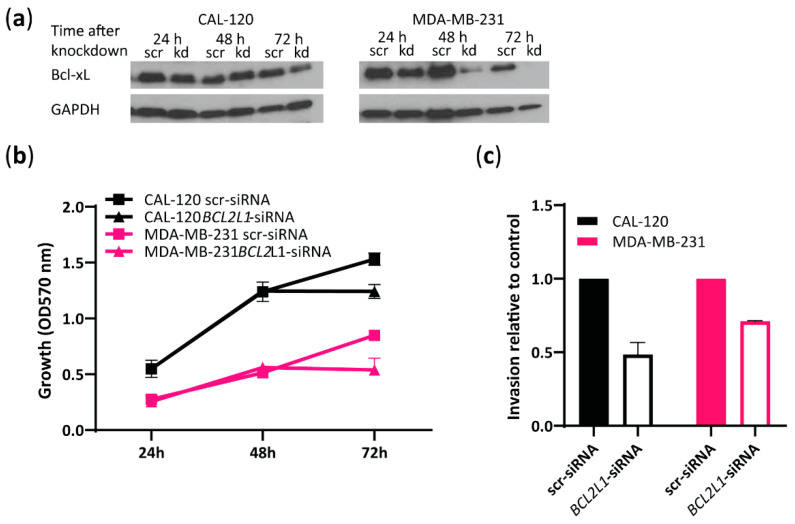
*BCL2L1* siRNA-mediated knockdown reduces the invasive properties of basal B TNBC cells. (**a**) Western blot analysis of Bcl-xL in CAL-120 and MDA-MB-231 cells before and after treatment with *BCL2L1*–siRNA or scrambled siRNA at 24, 48 and 72 h. GAPDH was used as loading control; (**b**) Evaluation of CAL-120 and MDA-MB-231 cell growth at 24, 48 and 72 h following *BCL2L1* siRNA-mediated knockdown determined by crystal violet colorimetric assay; (**c**) Analysis of invasive properties of CAL-120 and MDA-MB-231 cells at 72 h after treatment with *BCL2L1*–siRNA relative to cells treated with scrambled siRNA (control). The data represent the mean of two biological replicates ± SEM.

**Figure 5 ijms-23-10053-f005:**
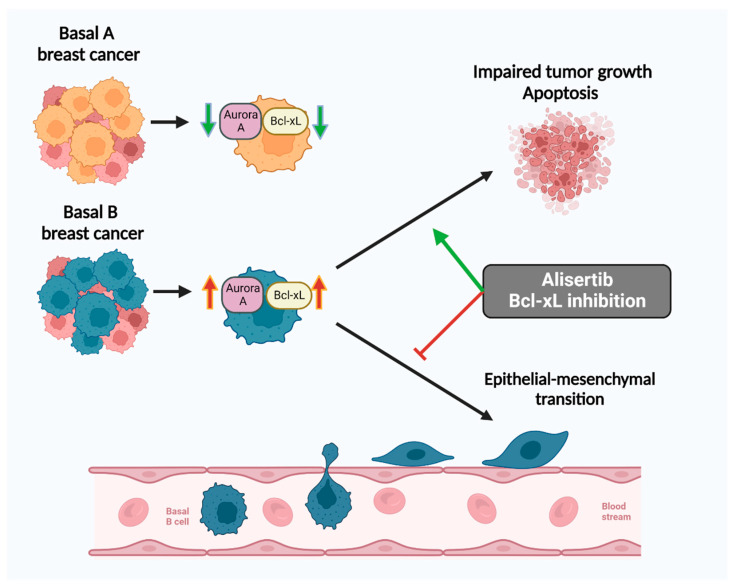
Inhibition of Aurora A with alisertib and blockage of Bcl-xL reduces invasion of basal B breast cancer cells. A schematic presentation showing the role of Aurora A and Bcl-xL in regulating basal B cell growth and metastatic capability (created with BioRender.com, Toronto, ON, Canada).

**Figure 6 ijms-23-10053-f006:**
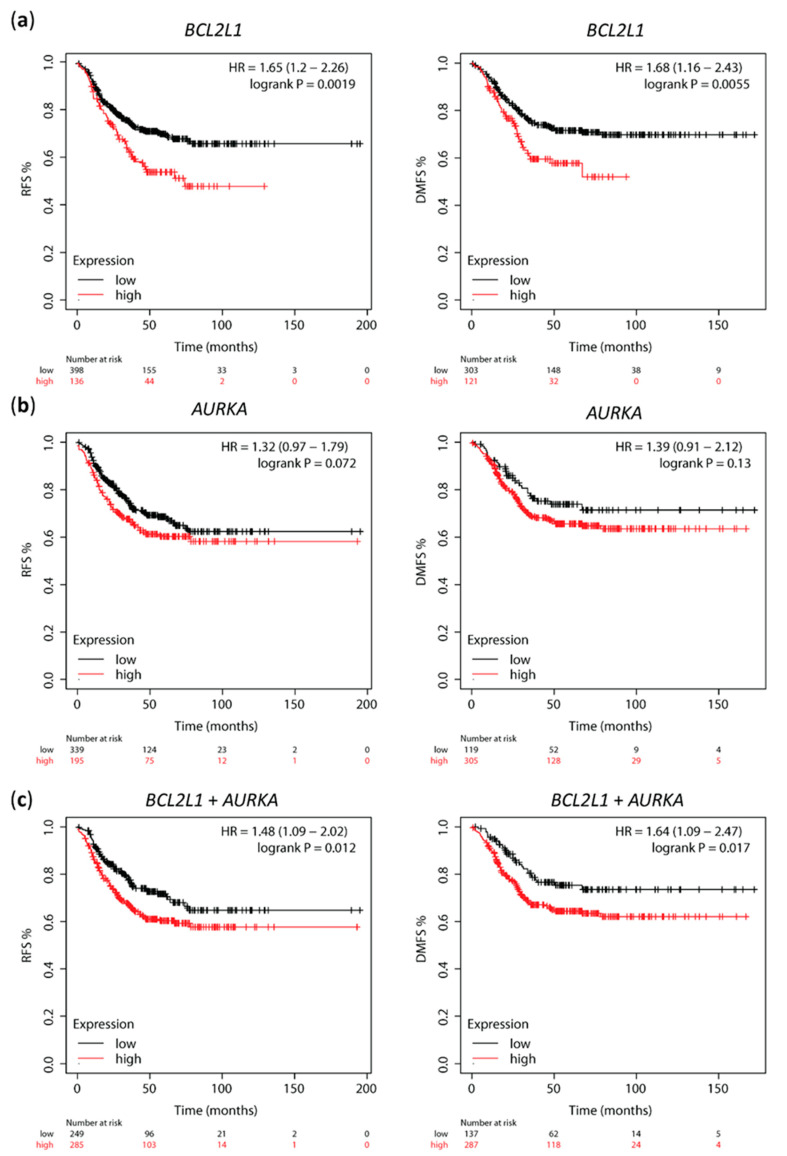
High *AURKA* and *BCL2L1* expression correlates with shorter relapse-free survival (RFS) and distant metastasis-free survival (DMFS) in TNBC. Kaplan–Meier survival curves of RFS and DMFS for (**a**) *BCL2L1* and (**b**) *AURKA* mRNA expression by KM plotter analysis; (**c**) Kaplan–Meier survival curves of RFS ad DMFS for combined *BCL2L1* and *AURKA* mRNA expression by KM plotter analysis.

**Table 1 ijms-23-10053-t001:** Multivariate analysis of RFS and DMFS according to *MKI67*, *ESR1*, *AURKA* and *BCL2L1* expression in TNBC tumors.

Variable	RFS HR (95% CI)	^1^ RFS *p*	DMFS HR (95% CI)	^1^ DMFS *p*
*MKI67*	0.89 (0.62–1.28)	0.5373	1.05 (0.72–1.53)	0.799
*ESR1*	1.42 (1.02–2)	0.0407 *	1.49 (1.02–2.19)	0.0412 *
*AURKA* + *BCL2L1*	1.65 (1.18–2.3)	0.0036 **	1.74 (1.12–2.69)	0.0134 *

^1^ RFS: Relapse-free survival; DMFS: Distant metastasis-free survival. * *p* ≤ 0.05 and ** *p* ≤ 0.01

## Data Availability

All datasets generated during the study will be made available upon reasonable request to the corresponding author, Henrik Ditzel, email address: hditzel@health.sdu.dk.

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
