# Peer review of "Aurora Kinase A and Bcl-xL Inhibition Suppresses Metastasis in Triple-Negative Breast Cancer"

_ijms, 2022, doi:10.3390/ijms231710053_

Round 1
Reviewer 1 Report
Dear Authors,
The manusiscript is well-written and the results are well-presented
I did not find it appropriate to not write the full name of the cell lines on the Y-axis such as (MB-231) instead of MDA-MB-231 cells (Not all readers are familiar with the full names of the cell lines)
Also, I would recommend a summative figure that highlights the findings of the paper
Author Response
Response to comments by reviewer #1:
The manuscript is well-written and the results are well-presented
Our response: We appreciate the reviewer’s positive comments.
I did not find it appropriate to not write the full name of the cell lines on the Y-axis such as (MB-231) instead of MDA-MB-231 cells (Not all readers are familiar with the full names of the cell lines)
Our response: As requested by the reviewer, we have now included the full name of the cell lines in the text and figures.
Also, I would recommend a summative figure that highlights the findings of the paper
Our response: As requested by the reviewer, we have now included a graphical abstract that summarizes the main findings of the paper, which was also included as new Fig. 5 in the main manuscript (p 10).
Reviewer 2 Report
Natascha and colleagues in this article have described how aurora Kinase A and Bcl-xL inhibition suppresses metastasis 2 in triple-negative breast cancer. Triple-negative breast cancer (TNBC) accounts for 10-15% of all breast cancer cases. TNBC represents an aggressive subtype of breast cancer that is associated with high metastatic ability, tumor heterogeneity and lack of effective therapies. Targeting the anti-apoptotic protein Bcl-xL together with anti-mitotic agents has recently been proposed as an efficient therapeutic strategy for different cancers. Breast cancer is the most common cancer in women worldwide, accounting for 31 %of all cancer cases. TNBC and the partly overlapping molecular subtype basal-like breast cancer represents an aggressive subtype of breast cancer. The authors showed gene amplification and protein upregulation of Aurora A and BclxL in basal B TNBC cell lines.
Chemical inhibition of Aurora A with alisertib and siRNA-mediated knockdown of BCL2L1 decreased the number of invading cells compared to non-treated cells in basal B cell lines.
Aspects of the results reported in this article potentially reinforce prior research in this field: “Treatment with alisertib induces increased expression of Bcl-xL in basal B TNBC cells, further supporting co-targeting both molecules,".
The article is suitable for publication with no further changes, as they showcase key findings,
1) High levels of AURKA either alone or together with BCL2L1 levels were not associated with altered patient survival.
2) Overexpression of Aurora A and Bcl-xL has been associated with breast cancer metastasis through induction of EMT.
3) Have examined the correlation between Aurora A and Bcl-xL expression and metastatic abilities in basal A and basal B TNBC, by confirming epithelial or mesenchymal features of MB-468, CAL-120 and MB-231 cells.
4) In agreement to the submitted article's findings, it was recently shown that combined inhibition of Aurora B and Bcl-xL caused synergistic cell-growth impairment in cancer cells.
5) Have clearly demonstrated how Aurora A and Bcl-xL both have specific impact on the ability of TNBC cells to invade and metastasize confirming its links to the metastatic process in different types of cancer, including breast cancer.
Author Response
Response to comments by reviewer #2:
Natascha and colleagues in this article have described how aurora Kinase A and Bcl-xL inhibition suppresses metastasis 2 in triple-negative breast cancer. Triple-negative breast cancer (TNBC) accounts for 10-15% of all breast cancer cases. TNBC represents an aggressive subtype of breast cancer that is associated with high metastatic ability, tumor heterogeneity and lack of effective therapies. Targeting the anti-apoptotic protein Bcl-xL together with anti-mitotic agents has recently been proposed as an efficient therapeutic strategy for different cancers. Breast cancer is the most common cancer in women worldwide, accounting for 31 %of all cancer cases. TNBC and the partly overlapping molecular subtype basal-like breast cancer represents an aggressive subtype of breast cancer. The authors showed gene amplification and protein upregulation of Aurora A and BclxL in basal B TNBC cell lines.
Chemical inhibition of Aurora A with alisertib and siRNA-mediated knockdown of BCL2L1 decreased the number of invading cells compared to non-treated cells in basal B cell lines.
Aspects of the results reported in this article potentially reinforce prior research in this field: “Treatment with alisertib induces increased expression of Bcl-xL in basal B TNBC cells, further supporting co-targeting both molecules,".
The article is suitable for publication with no further changes, as they showcase key findings,
1) High levels of AURKA either alone or together with BCL2L1 levels were not associated with altered patient survival.
2) Overexpression of Aurora A and Bcl-xL has been associated with breast cancer metastasis through induction of EMT.
3) Have examined the correlation between Aurora A and Bcl-xL expression and metastatic abilities in basal A and basal B TNBC, by confirming epithelial or mesenchymal features of MB-468, CAL-120 and MB-231 cells.
4) In agreement to the submitted article's findings, it was recently shown that combined inhibition of Aurora B and Bcl-xL caused synergistic cell-growth impairment in cancer cells.
5) Have clearly demonstrated how Aurora A and Bcl-xL both have specific impact on the ability of TNBC cells to invade and metastasize confirming its links to the metastatic process in different types of cancer, including breast cancer.
Our response: We appreciate the reviewer’s positive comments.
Reviewer 3 Report
The work investigates the role of amplification and overexpression of Aurora A and Bcl-xL in two mesenchyme-like basal B cell lines compared to the epithelial-like breast cancer A basal cell line. The outcomes of these researches imply that targeting Aurora A and Bcl-xL to inhibit tumor metastasis in the TNBC primary B subgroup may be a useful therapeutic strategy.
The manuscript represents an interesting study and sufficiently contributes to the scientific understanding of treating triple-negative breast cancer. It is clear, well organized, and well written. Moreover, the well-documented results of these studies are compiled in Supplementary Materials. I have no criticism of this manuscript. I think this is a good job.
The disadvantages of this work are as follows:
Since alisertib was used in the research, the Authors should provide more space to the characteristics of this substance. They should provide, among others, systematic name, chemical formula and basic properties of alisertib as an Aurora kinase inhibitor.
Abbreviations and acronyms must be defined on first use, therefore the abbreviations must be defined PARP, TE, SDS, PVDF and others.
References are not structured in terms of the IJMS format. Some journal names were abbreviated, others were not. Many journal names are written in lowercase.
Author Response
Response to comments by reviewer #3:
The work investigates the role of amplification and overexpression of Aurora A and Bcl-xL in two mesenchyme-like basal B cell lines compared to the epithelial-like breast cancer A basal cell line. The outcomes of these researches imply that targeting Aurora A and Bcl-xL to inhibit tumor metastasis in the TNBC primary B subgroup may be a useful therapeutic strategy.
The manuscript represents an interesting study and sufficiently contributes to the scientific understanding of treating triple-negative breast cancer. It is clear, well organized, and well written. Moreover, the well-documented results of these studies are compiled in Supplementary Materials. I have no criticism of this manuscript. I think this is a good job.
Our response: We appreciate the reviewer’s positive comments.
The disadvantages of this work are as follows:
Since alisertib was used in the research, the Authors should provide more space to the characteristics of this substance. They should provide, among others, systematic name, chemical formula and basic properties of alisertib as an Aurora kinase inhibitor.
Our response: As requested by the reviewer, we have now included more information on alisertib, such as name, chemical formula and basic properties in the Materials and Methods section (p 14).
Abbreviations and acronyms must be defined on first use, therefore the abbreviations must be defined PARP, TE, SDS, PVDF and others.
Our response: As the reviewer rightly points out, abbreviations and acronyms must be defined on first use, therefore we have now included definitions of all abbreviations.
References are not structured in terms of the IJMS format. Some journal names were abbreviated, others were not. Many journal names are written in lowercase.
Our response: As the reviewer rightly points out, references were not complying with IJMS format. This has now been corrected.